# Motif-Targeting Phosphoproteome Analysis of Cancer Cells for Profiling Kinase Inhibitors

**DOI:** 10.3390/cancers15010078

**Published:** 2022-12-23

**Authors:** Kosuke Ogata, Shunsuke Takagi, Naoyuki Sugiyama, Yasushi Ishihama

**Affiliations:** Graduate School of Pharmaceutical Sciences, Kyoto University, Kyoto 606-8501, Japan

**Keywords:** proteomics, phosphoproteomics, motif-targeting, protein kinase, molecular-targeting drug

## Abstract

**Simple Summary:**

Phosphoproteomics is essential for basic understanding of cell biology. However, analysis of the phosphoproteome remains far from comprehensive. Therefore, we proposed a new method to enrich a subset of the phosphoproteome and expand the scope of analysis by using in vitro kinase reactions. We showed our novel workflow identified and quantified the phosphopeptides which have not been observed in the conventional workflow. We also demonstrated that this method is effective for profiling kinase inhibitors. We envision our workflow could easily be adapted to target different subsets of the phosphoproteome by utilizing different sets of kinases, and should be applicable to variety of samples.

**Abstract:**

We present a motif-targeting phosphoproteome analysis workflow utilizing in vitro kinase reaction to enrich a subset of peptides with specific primary sequence motifs. Phosphopeptides are enriched and dephosphorylated with alkaline phosphatase, followed by in vitro kinase reaction to phosphorylate substrate peptides with specific primary-sequence motifs. These phosphopeptides are enriched again, TMT-labeled, dephosphorylated to enhance MS-detectability, and analyzed by LC/MS/MS. We applied this approach to inhibitor-treated cancer cells, and successfully profiled the inhibitory spectra of multiple kinase inhibitors. We anticipate this approach will be applicable to target specific subsets of the phosphoproteome using the wide variety of available recombinant protein kinases.

## 1. Introduction

The biological function and structure of eukaryotic proteins are regulated by many kinds of post-translational modifications (PTMs). Reversible protein phosphorylation functions as a molecular switch for intracellular signaling and is one of the most important modifications. It plays a pivotal role in almost all biological processes, including cell division, differentiation, polarization, and apoptosis [1,2]. Recent advances in mass spectrometry (MS)-based proteomics, together with highly selective phosphopeptide enrichment methods utilizing metal affinity chromatography with immobilized metal ion or titanium dioxide beads, have made it possible to identify tens of thousands of phosphorylation sites from cell and tissue samples in a single study [3,4,5,6]. However, the identification of low-abundance phosphosites is still a major challenge in current LC/MS/MS approaches because the co-existing high-abundance species suppress the electrospray ionization of low-abundance ones in general. Therefore, it is important to ensure that proteomic samples do not exceed the capabilities of the LC/MS/MS system in both peptide number and dynamic range [5].

One of the commonly used techniques to increase the coverage of the proteome is sample fractionation [7]. Fractionation reduces the complexity of the sample in a single LC/MS analysis and can enable identification of low-abundance peptides. It can also be applied in phosphoproteomics to identify low-abundance phosphorylation sites. However, it requires a larger input sample amount due to the lower recovery of the peptide after fractionation, and a substantially longer instrument time to analyze each sample, making large-scale analysis difficult. Another way to identify low-abundance phosphopeptides is to selectively enrich a part of the phosphopeptides to create phosphoproteomic subsets. For example, selective enrichment of mono-phosphorylated and multi-phosphorylated peptides improves the coverage by reducing the complexity of the samples, owing to a decrease in ionization suppression of multi-phosphorylated peptides having more negative charges [8]. Selective enrichment of pTyr (phosphotyrosine) peptides can be used to identify more pTyr sites [9,10,11]. Chemical enrichment of cysteine-containing peptides enables identification of the cys-containing phosphoproteome subset [12]. Context-independent motif-specific antibodies directed against short amino acid motifs around the phosphosites can also be used to enrich the specific motif-containing phosphopeptides from the phosphoproteome [13,14]. However, it is not possible to target all types of phosphopeptides in this way, due to limitations of available probes or antibodies.

Protein phosphorylation is mediated by protein kinases, which have substrate preferences. We previously conducted an LC/MS/MS-based in vitro kinase assay to profile human protein kinome using dephosphorylated lysate proteins as the substrate source for in vitro kinase reactions [15]. A total of 175,574 direct kinase substrates were identified for 385 protein kinases. By using these in vitro kinase reactions in combination with phosphopeptide enrichment technology, the specific subset of peptides recognized by a particular kinase can be enriched from complex proteome samples. Tsai et al. previously developed a method that utilizes in vitro kinase reactions with CK2, MAPK, and EGFR to generate phosphopeptides with targeted motifs from non-phosphorylated peptides, enabling measurement of the phosphorylation stoichiometry of more than 1000 phosphorylation sites [16].

Other than specific enrichment of subsets of the phosphoproteome, a technique called tandem mass tag (TMT) boosting has also been reported to identify low abundance phosphopeptides; this facilitates target peptide detection not by enrichment, but by the addition of target peptides to one of the TMT channels to boost the signals of their precursor ions [17,18,19]. We previously reported a motif-centric quantitation approach that utilizes the combination of in vitro kinase reaction and this TMT-boosting strategy, enabling low-abundance phosphopeptides to be quantified from limited amounts of samples [20]. However, this strategy does not reduce the complexity of the phosphoproteome samples. The complex nature of the samples prevents the accurate quantitation of low-abundance signals due to contamination with non-target peptide precursor ions, requiring additional separation techniques and/or an instrument capable of tandem MS3 experiments [21,22,23,24].

Here, we propose a phosphoproteomic subset approach in which phosphoproteome samples after phosphopeptide enrichment are completely dephosphorylated and then selectively phosphorylated with specific kinases. We call a phosphoproteome subset obtained in this way the motif-targeting phosphoproteome, because the kinases phosphorylate peptides containing the specific substrate sequence motifs. In the present study, the tyrosine kinase-based motif-targeting approach identified over three times more tyrosine-containing phosphopeptides than the conventional global phosphoproteomic approach. We also applied this new method to the profiling of kinase inhibitors in order to establish their inhibitory activity spectra. We anticipate this workflow will be useful to target specific sets of the phosphoproteome by utilizing the wide variety of available recombinant protein kinases.

## 2. Materials and Methods

### 2.1. Materials

Titanium dioxide (TiO_2_, titania) particles (Titansphere, particle size: 10 μm), Empore SDB-XC and C8 membrane disks were obtained from GL Sciences (Tokyo, Japan). Sequencing-grade modified trypsin was purchased from Promega (Madison, WI, USA). Recombinant protein kinases were obtained from Carna Biosciences (Kobe, Japan). Kinase inhibitors were purchased from Selleck Chemicals (Houston, TX, USA). Water was purified by a Millipore Milli-Q system (Bedford, MA, USA). Alkaline phosphatase from calf intestine, DL-lactic acid, MS-grade Lys-C (lysyl endopeptidase), and all other chemicals were purchased from Fujifilm Wako (Osaka, Japan).

### 2.2. Preparation of HeLa Cells

HeLa S3 cells from Japan Health Sciences Foundation were cultured to 80% confluence with DMEM containing 10% FBS in 15 cm diameter dishes. Cells were rinsed twice with ice-cold PBS, scraped, collected and pelleted by centrifugation. For inhibitor treatments, the cells that reached 80% confluence were incubated with DMEM containing 0.1% FBS for 24 h. The cells were treated with inhibitors dissolved in dimethyl sulfoxide (DMSO) (10 μM final concentration) or vehicle DMSO for 30 min. After treatment, the medium was replaced with DMEM/10% FBS and incubation was continued for 15 min. Cells were scraped, collected and pelleted by centrifugation.

The cell pellets were lysed with 1 mL of lysis buffer (12 mM sodium deoxycholate, 12 mM sodium *N*-lauroylsarcosinate in 100 mM Tris-HCl, pH 9.0 containing protease inhibitors (Sigma), protein phosphatase inhibitor cocktail 2 and 3 (Sigma)) [25]. The lysates were heated at 95 °C for 5 min and then sonicated for 20 min on ice. The extracted proteins were quantified with a BCA Protein Assay Kit. Proteins were reduced with 10 mM dithiothreitol for 30 min, and alkylated with 50 mM iodoacetamide for 30 min in the dark. After that, the samples were diluted with 4 mL of 50 mM ammonium bicarbonate, and digested for 3 h with Lys-C and for 16 h with trypsin at room temperature. After digestion, the samples were mixed with 1 mL of ethyl acetate, and acidified by the addition of 20% trifluoroacetic acid (TFA) (final concentration: 0.5%). The aqueous and organic phases were completely separated by centrifugation at 15,800 g for 2 min. The aqueous phase was collected and desalted using StageTips with SDB-XC Empore disk membranes [26,27].

### 2.3. Phosphopeptide Enrichment

C8 StageTips packed with Titansphere (Metal oxide chromatographic (MOC) tips, 0.5 mg/100 μg of digests) [28] were equilibrated with 80% acetonitrile (ACN) containing 300 mg/mL lactic acid and 0.1% TFA (solution A). The digested samples (500 μg/500 μL) were diluted with 500 μL of solution A and loaded onto the MOC tips. MOC tips were washed with solution A and 80% ACN containing 0.1% TFA. Phosphopeptides were eluted with 0.5% piperidine, acidified immediately by adding 10% TFA (final concentration: 0.1%), and desalted using SDB-XC StageTips.

### 2.4. Dephosphorylation of Phosphopeptides

Enriched phosphopeptides were dissolved in 25 μL of 100 mM Tris-HCl buffer (pH 9.0). Alkaline phosphatase (5 units) was added to the solution and the mixture was incubated for 3 h at 37 °C. After the reaction, the buffer was acidified by adding 10% TFA 1.25 μL. The samples were desalted using StageTips packed with SDB-XC membrane.

### 2.5. In Vitro Kinase Reactions

The dephosphorylated peptides were dissolved in 10 μL of 40 mM Tris-HCl buffer (pH 7.5) with 1 mM ATP and 20 mM MgCl_2_. After addition of the kinases (100 ng FYN, JAK3 and MER), the samples were incubated at 37 °C for 3 h. After the reaction, the buffer was acidified by adding 10% TFA 1 μL. The samples were desalted using StageTips packed with SDB-XC membrane. Phosphopeptides were enriched again using MOC tips as described above.

### 2.6. TMT Labeling

The desalted phosphopeptide samples were dried and resuspended in 5 μL of 200 mM 4-(2-hydroxyethyl)-1-piperazineethanesulfonic acid (HEPES) pH 8.5. After that, TMT10plex label reagent (25 μg) dissolved in 5 μL of ACN was added to the sample. The mixture was incubated for 1 h at room temperature. Hydroxylamine (final concentration: 0.33%) was added to the mixture to quench the reaction. The samples were acidified, diluted to make the ACN concentration less than 5%, and desalted using SDB-XC StageTips [29].

TMT-labeled phosphopeptides were further dephosphorylated with 5 units of alkaline phosphatase. The reaction conditions were the same as described above. After the reaction, the buffer was acidified by adding 10% TFA 1.25 μL. The samples were desalted using SDB-XC StageTips.

### 2.7. NanoLC/MS/MS Analyses

NanoLC/MS/MS analyses were performed on a Q Exactive mass spectrometer (Thermo Fisher Scientific, Waltham, MA, USA) equipped with an UltiMate 3000 RSLCnano pump (Thermo Fisher Scientific) and a PALxt HTC autosampler (CTC Analytics, Zwingen, Switzerland). A 100 μm inner diameter column (15 cm length) in-house-packed with 3 μm reversed-phase silica beads (ReproSil-Pur C18-AQ, Dr. Maisch, Ammerbuch, Germany) was used for the analysis. 5 μL of peptide solution was injected and separated with a 65 min linear gradient from 5% to 40% B at the flow rate of 500 nL/min. The mobile phases consisted of 0.5% acetic acid (A) and 0.5% acetic acid and 80% acetonitrile (B) were used. The spray voltage of 2.4 kV and the heated capillary temperature of 240 °C were used. In each MS scan, 10 precursor ions were selected for subsequent MS/MS scans. Full scan resolution was set to 70,000 at *m*/*z* 200. The mass range was set to 300–1500. The resolution and AGC target for MS2 scans were set at 35,000 and 1 × 10^5^, respectively. Isolation width was set at 1.4 Th. The normalized collision energy was set at 33. Precursor ions with charge states 2 to 5 were targeted for MS2. Data was collected with profile mode. The lock mass function was enabled.

### 2.8. Database Searching and Data Analysis

Peptides and proteins were identified by means of automated database searching using MaxQuant v2.0.1.0 [30,31] against UniprotKB/Swissprot release 2022_02 with Trypsin/P specificity allowing for up to 2 missed cleavages. TMT6plex (N-Term), TMT6plex (K) and Carbamidomethyl (C) were set as fixed modifications. Acetyl (Protein N-term) and Oxidation (M) were allowed as variable modifications. 1% false discovery rate (FDR) cutoffs (peptide spectrum match (PSM) and protein) were employed for identifications.

The search result was processed with the Perseus 1.6.14.0 [31] software for the quantitative analysis. For kinase inhibitor profiling, the normalization of reporter ion intensities was performed by the median subtraction on Perseus. First, the reporter intensity values of each channel were subtracted by median, and then the reporter intensity values of each spectrum were subtracted by median. Resulting table is used for calculating the correlation among samples, clustering and creating heatmap. Clustering analysis was performed by Python SciPy (1.4.1) library. The significantly regulated peptides were extracted by multiple-sample tests function implemented in Perseus, with following parameters: Test: ANOVA, S0: 0, Permutation-based FDR, FDR: 0.05.

## 3. Results and Discussion

One of the major barriers to comprehensive phosphoproteomics is the large dynamic range of the phosphoproteome. In general, LC/MS/MS has an identification bias toward the more abundant species or species with high ionization efficiency in a sample. Phosphopeptides of low-abundance proteins and low stoichiometry phosphosites are difficult to quantify because of the presence of the high-abundance (phospho)peptides, even after selective enrichment of phosphopeptides which exclude non-phosphopeptides from samples. Here, we first performed a global phosphoproteome analysis of HeLa cells using a conventional phosphopeptide enrichment approach with a combination of single-step TiO_2_ enrichment and TMT labeling. Four replicate LC/MS/MS analyses (process replicate) of the enriched phosphopeptides led to the identification of 2118 phosphopeptides reproducibly (at least 3 out of 4 runs). It was found that 2078/1087/419 peptides contained at least one S/T/Y residue, respectively, reflecting the higher abundance of S/T phosphorylation compared with Y phosphorylation in human cells. Thus, we hypothesized that the selective enrichment of tyrosine-containing phosphopeptides (Tyr-phosphoproteome subset) would increase the coverage.

To enrich the tyrosine-containing peptides in the human phosphoproteome, we utilized in vitro kinase reaction to attach a phosphorylation “tag” to the target phosphopeptides, because phosphopeptides are easily enriched by metal affinity chromatography. To achieve this, we first enriched phosphopeptides from HeLa cells by TiO_2_ metal oxide chromatography, then completely dephosphorylated them by the addition of alkaline phosphatase and conducted in vitro kinase reaction to phosphorylate the desired subset of peptides. The resulting phosphopeptides were enriched again, TMT-labeled, and dephosphorylated to enhance the MS detectability (Figure 1). The dephosphorylation of phosphopeptides increase the detectability of peptide sequence and lead to higher coverage of phosphoproteome [32,33,34].

Since this workflow dephosphorylates the originally phosphorylated peptides, the original phosphorylation information is lost. To ensure that the identified peptides were indeed originally phosphorylated, we needed to confirm that the sample after the first TiO_2_ enrichment only contains phosphopeptides, because contaminating non-phosphopeptides would perturb the subsequent quantitative analysis. We analyzed the phospho-enriched samples and found that over 98% of identified peptides were phosphorylated (Appendix A). This indicates that almost all the identified peptides originated from a phosphopeptide, so that the subsequent dephosphorylation process would not materially affect the quantitation of the phosphopeptides. Furthermore, we evaluated the dephosphorylation efficiency of the alkaline phosphatase treatment and confirmed that more than 96% of phosphopeptides were successfully dephosphorylated, so that contamination with non-targeted phosphopeptides is minimal (Appendix A). Next, to select tyrosine kinases suitable for the in vitro kinase reaction, we screened the kinases registered in the Kinapple database [15], which contains in vitro kinase substrate information. We selected 12 tyrosine kinases based on the number of phosphosites identified and the differences of selectivity in order to cover more tyrosine phosphopeptide primary sequence motifs. We tested each kinase individually with our workflow and found that FYN, JAK3, and MER provided the largest numbers of tyrosine-phosphorylated peptides (Appendix A). A cocktail of these three kinases provided even more tyrosine-phosphorylated peptides (Appendix A), and thus we employed a mixture of these three kinases for our workflow. Of the identified phosphotyrosine sites with FYN, JAK3 or MER (1305 sites), 71% (935 sites) were already reported in PhosphoSitePlus database (Appendix A).

After establishing the workflow, we profiled the peptides identified by means of this strategy. We could identify 1341 peptides reproducibly (at least 3 out of 4 runs, process replicate), of which 89.9% (1206) contained at least one tyrosine residue in the peptide sequence. For comparison, the phospho-enriched and subsequently dephosphorylated sample (conventional method) was analyzed under the same LC-MS and data analysis conditions. Of the 1987 peptides identified, 19.6% (389) contained tyrosine, indicating that our motif-targeting approach could identify 3.4 times more tyrosine-containing peptides (Figure 2A and Appendix A). Overlap of tyrosine-containing peptides showed that the present workflow covered almost 85 % of the tyrosine-containing peptides identified in the conventional sample, indicating that the tyrosine-targeting workflow identifies more tyrosine-containing peptides while maintaining high coverage of the peptides identified with the conventional workflow (Figure 2B). Another potential benefit of the motif-targeting workflow is the reduction of the sample complexity, resulting in the less contamination of the precursor ions that leads to the better quantitative accuracy of isobaric-tag based experiment. TMT-based quantitation often suffers from contamination of non-target precursor ions in the MS2 spectra, leading to reduced quantitation accuracy (so-called “ratio compression”). To improve the quantitation results, better separation of peptides by sample fractionation and/or ion mobility spectrometry, or MS3-based purification of the fragment ions is needed [21,22,23,24]. We hypothesized that the motif-targeting workflow would reduce the complexity of the sample because it targets a specific subset of phosphopeptides in the sample. To confirm this, we checked the LC/MS spectrograms (*m/z*-retention time plots). Compared with the conventional runs, the raw data became sparser and fewer peaks were observed in the motif-targeting runs (Figure 3A,B). The number of peptide ion-like features detected in MaxQuant [30] was also decreased (241,486 for 4 conventional runs, 196,367 for 4 motif-targeting runs). These results confirm that the motif-targeting approach reduces the number of peptide species in the sample. Furthermore, we profiled the Precursor Ion Fraction (PIF) of the commonly identified peptides between conventional runs and motif-targeting runs; this is available in MaxQuant [30] and indicates the purity of the precursor ions in the given peptide spectrum match. The ratio of high-purity matches (PIF > 90%) in the commonly identified peptides was increased from 70% (conventional) to 82% (motif-targeting) with the isolation window of 1.4 Th, suggesting that the TMT ratio compression effect is less significant in the motif-targeting runs (Figure 3C and Appendix A).

Finally, we profiled kinase inhibitors by means of the phosphoproteomic profiling of kinase inhibitor-treated HeLa cells, using the established motif-targeting workflow. Protein kinases have emerged as major drug targets in cancer therapy because of their central role in the signaling cascades. ATP-binding sites within the kinase domain are targeted by these molecules, and these sites are highly conserved among the 518 human protein kinases. This makes it difficult to develop drugs that are highly selective for specific proteins, and most kinase inhibitors possess off-target as well as on-target effects on their intended targets. This can broaden the therapeutic spectrum of the drug, but can cause some undesired side effects [35]. We selected 8 inhibitors (afatinib, selumetinib, silmitasertib, vemurafenib, crizotinib, ponatinib, sunitinib, and tofacitinib), which cover various target pathways, though they are also known to inhibit considerable numbers of off-target kinases [36,37].

We labeled the inhibitor-treated samples with TMT10plex (2 vehicle DMSO-treated and 8 inhibitor-treated) (Figure 4A). From the duplicate inhibitor treatment experiments, we quantified 2203 peptides and 1519 peptides in conventional and motif-targeting runs, respectively, both in duplicate. The numbers of tyrosine containing peptides are 428 and 1358, respectively. The log-normalized intensities in each channel were compared with each other. The overall correlation of quantitative results between channels of both experimental sets is shown in heat maps (Figure 4B,C). The replicate samples with the same inhibitor treatment were clustered together, indicating that the motif-targeting workflow enabled reproducible quantification of peptides despite the many sample processing steps in this workflow. Though the clustering results indicated good reproducibility of the workflow, the clusters appeared slightly different from each other. In conventional runs, the afatinib- and selumetinib-treated samples clustered near the control DMSO groups (Figure 4B and Appendix A). On the other hand, in motif-targeting runs, these samples were well differentiated from the DMSO groups, and the closest cluster to the DMSO groups was the silmitasertib-treated group, which was well differentiated from DMSO groups in conventional runs (Figure 4C and Appendix A). This is probably due to the characteristics of these inhibitors. Silmitasertib targets CK2 in a highly specific manner [36], while CK2 phosphorylates serine adjacent to acidic residues. Our results may indicate that neither CK2 nor its downstream pathways target phosphorylation on or near tyrosine residues. Afatinib and selumetinib both target EGFR signaling pathways in which tyrosine kinases and MAP kinases play central roles. These pathways might be more relevant to phosphorylation around tyrosine residues. Also, the co-clustering of these two inhibitors suggests that they have similar inhibitory effects on the EGFR pathway, although these inhibitors target this pathway differently.

To profile what kinds of substrate phosphorylation changes were induced by inhibitor treatments, we combined the identified peptides from both experimental sets. The peptide overlap between these two data sets is only 15% (484 peptides overlapped between the two data sets), and a total of 3238 peptide quantitative data were obtained. The significantly changed peptides were extracted by using ANOVA significance analysis of log-normalized intensities with the module in Perseus software (version 1.6.14.0), identifying 1409 peptides that were regulated by at least one kinase inhibitor treatment. We visualized the regulated peptides as a function of treatment by hierarchical clustering of z-score-normalized intensities, which grouped regulated substrates according to the inhibitor treatment (Figure 5A and Appendix A). The conventional and motif-targeting data sets reveal different sets of peptides significantly regulated by the inhibitor treatments (Figure 5B). Notably, the 453 out of 578 (78%) significant hits which uniquely detected in motif-targeting workflow were even not identified in the conventional workflow. Furthermore, of the remaining 125 peptides which were also identified in the conventional group, 82 peptides have a higher identification score than those in the conventional group. These results indicated, by the target extraction of the phosphopeptide of interest, the motif-targeting approach can extend the coverage of the phosphoproteome. The clustering analysis showed ten distinct groups of peptides which responded similarly to the inhibitor treatment. The peptide SVSSYGNIR from Raptor, which contains the S721 phosphorylation site that is directly phosphorylated by RSK1 [38], appears in cluster 3, which showed downregulation in a sunitinib treatment-specific manner. The z-score of this peptide was compared with previously reported %-inhibition values of recombinant RSK1 [36] for 6 out of 8 inhibitors evaluated here. Interestingly, the z-scores and %-inhibition values were highly correlated (R^2^ = 0.876), supporting the idea that direct inhibition of the upstream RSK1 was the main contributor to the downregulation of this peptide. We also found that the peptide VADPDHDHTGFLTEYVATR from ERK2, which contains well-known downstream phosphorylation sites of the EGFR pathway (T185 and Y187), showed a significant decrease in the afatinib- and selumetinib-treated samples (Cluster 2). In addition, HTDDEMTGYVATR from p38α showed a significant decrease in a tofacitinib (JAK inhibitor)-specific manner (Cluster 9), reflecting the close relationship of the JAK and MAPK pathways including p38α [39,40,41]. These peptides were only identified by means of the motif-targeting approach in this study, demonstrating the usefulness of this workflow.

## 4. Conclusions

Identifying a wide range of phosphopeptides is indispensable for biological research, since important signals may be transduced via specific kinases that are expressed at extremely low levels. We have established a phosphoproteomic subset approach, called the motif-targeting method, which targets a specific set of phosphopeptides based on the substrate recognition preferences of recombinant kinases. We showed that a tyrosine kinase-based motif-targeting approach made the sample less complex, facilitating quantitative analysis, and enabled the identification of 3.4 times more tyrosine-containing peptides than the conventional workflow.

The present workflow requires dephosphorylation of the originally phosphorylated peptides. Previously, it was reported that the dephosphorylation of phosphopeptides improves their detectability, leading to better coverage of the phosphoproteome [32,33,34]. However, it is necessary to carry out additional validation to unambiguously determine phosphosite locations, because modification localization information is lost during dephosphorylation. Furthermore, quantification of individual phospho isoforms, which have the same peptide sequences but different phospho-modification sites, is not possible because, after dephosphorylation, only the sum of all phospho isoforms is quantified. Further validation study utilizing a targeted proteomics approach will be necessary to evaluate candidate peptides identified in this study. Nevertheless, this motif-targeting approach is effective in extending the coverage of the phosphoproteome, and was also successfully applied to profile the activity spectra of kinase inhibitors. We anticipate that this workflow could easily be adapted to target different subsets of the phosphoproteome by utilizing different sets of kinases, and should be applicable to many kinds of samples.

## Figures and Tables

**Figure 1 cancers-15-00078-f001:**
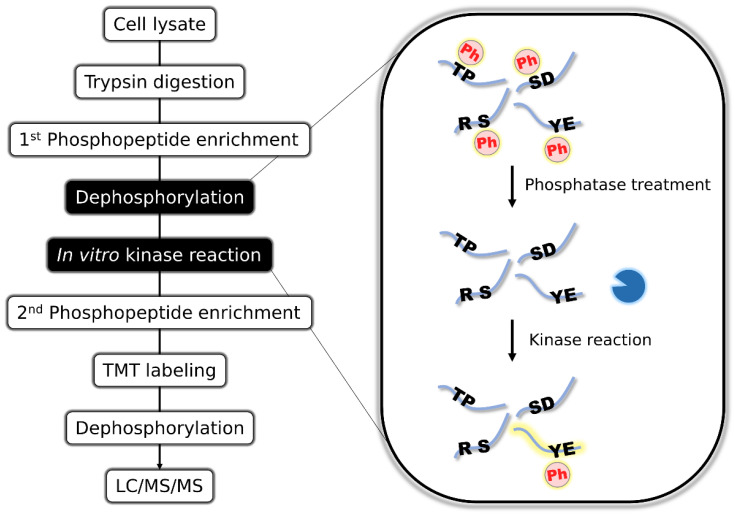
Schematic representation of the motif-targeting workflow. Phosphopeptides are first enriched from cell lysates by TiO_2_ metal oxide chromatography. Enriched phosphopeptides are completely dephosphorylated by the addition of alkaline phosphatase, followed by in vitro kinase reaction to phosphorylate the peptides with specific primary sequence motifs. The derived phosphopeptides are enriched again, TMT-labeled, dephosphorylated to enhance the MS detectability, and analyzed by LC/MS/MS.

**Figure 2 cancers-15-00078-f002:**
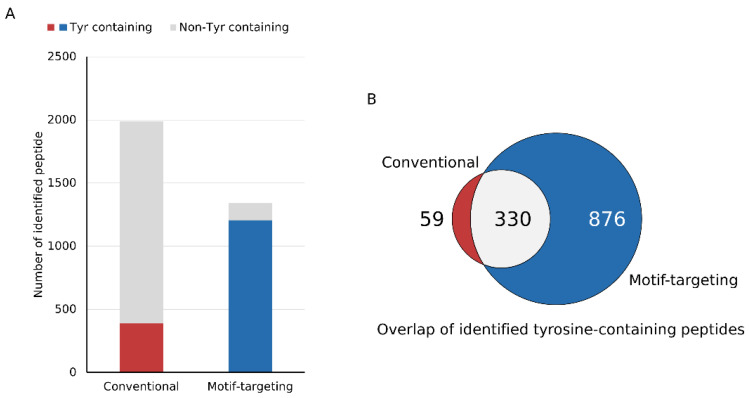
Identification results of tyrosine-containing peptides. (**A**) The bar chart shows the identified number of peptides in each experimental workflow. The peptides are categorized based on the presence of tyrosine residues. (**B**) The Venn diagram shows the number of tyrosine-containing peptides identified in each set. The motif-targeting workflow identified almost 85% (330 out of 389) of the peptides found in the conventional workflow.

**Figure 3 cancers-15-00078-f003:**
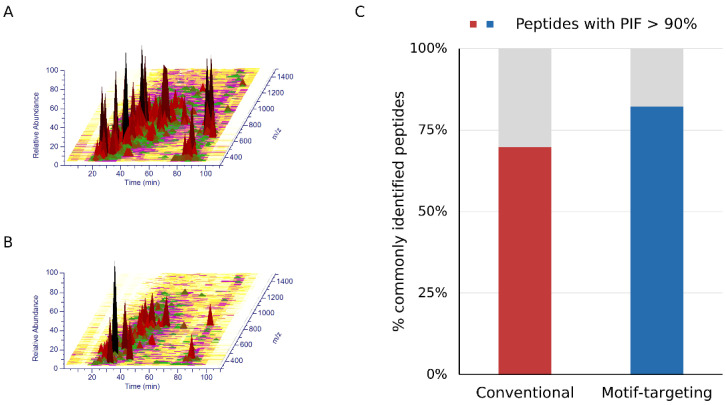
The complexity of the samples. A, B. LC/MS spectrograms obtained by conventional (**A**) and motif-targeting (**B**) workflows. The peaks are colored by their abundances. (**C**) The bar chart shows the ratios of high-purity matches (PIF > 90%) among the peptides commonly identified by the conventional and motif-targeting workflows.

**Figure 4 cancers-15-00078-f004:**
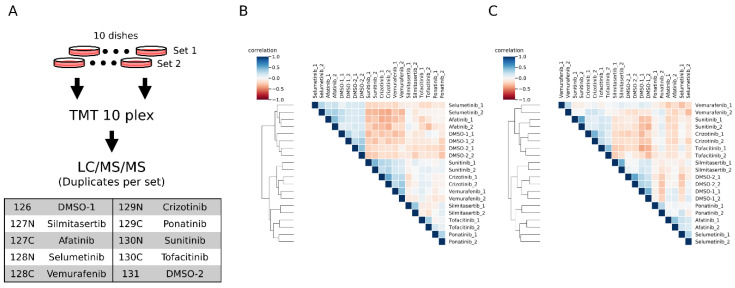
Phosphoproteomic profiling of inhibitor-treated HeLa cells. (**A**) Schematic representation of the workflow. 10 cell dishes were prepared for each set of experiments, and treated with different inhibitors or vehicle DMSO. The peptides extracted from each cell dish were labeled with each channel of TMT 10 plex labeling reagents shown in the table. (**B**,**C**) Heatmaps show the unsupervised clustering results of Kendall correlations calculated with the normalized intensities. The results from conventional (**B**) and motif-targeting (**C**) workflows are shown.

**Figure 5 cancers-15-00078-f005:**
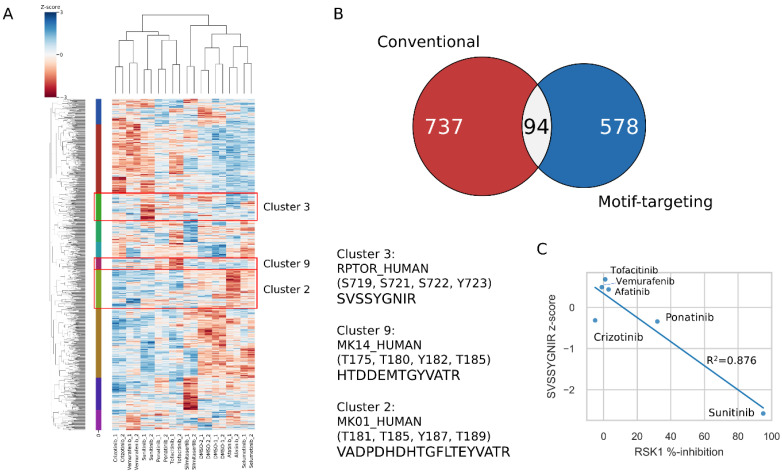
Clustering analysis of significantly changed peptides. (**A**) The heatmap shows the clustering results for significantly changed peptides. The peptides are clustered into ten groups. Several candidate peptides representing each cluster are shown. (**B**) The overlap of significantly regulated peptides identified in the conventional workflow and motif-targeting workflow. (**C**) The relationship between %-inhibition of RSK1 [36] and the z-scores from the quantitative results.

## Data Availability

The MS raw data and analysis files including Perseus processed file have been deposited with the ProteomeXchange Consortium (http://proteomecentral.proteomexchange.org) via the jPOST partner repository [42,43] (https://jpostdb.org) with the data set identifier PXD037644, accessed on 23 December 2022.

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
