# Peer review of "Motif-Targeting Phosphoproteome Analysis of Cancer Cells for Profiling Kinase Inhibitors"

_cancers, 2022, doi:10.3390/cancers15010078_

Round 1
Reviewer 1 Report
In this manuscript, the authors modified their previous workflow of in-vitro kinase reactions and TMT labeling to profile phosphorated peptides, and applied the new workflow to characterize phosphoproteome regulated by kinase inhibitors. Overall the manuscript is well written. There are still some concerns that the authors need to address before the manuscript can be considered for publication.
1. Line 26-28: Limited loading capacity results from hardware, not a problem of shotgun proteomics. The authors should change the statement. Also, there should be some transition sentences between “identifying low abundance proteins” and “shotgun proteomics”.
2. Line 49: Please correct kinomes to kinome.
3. Line 58-61: I don’t understand the relationship between TMT labeling and enrichment. I think the authors want to highlight the advantages of TMT labeling, e.g., enhanced ionization by TMT tagging and peptide identification by multiplexing. The authors should change the statement to make it clear to readers.
4. Line 175: Please change greater to higher.
5. Line 183: The authors should provide data here to support “completely dephosphorylated”.
6. Line 198-199: It’s good that the authors verified the enrichment efficiency of the TiO2 approach and the dephosphorylation efficiency of the alkaline phosphatase reaction to minimize the contamination of non-phosphorated peptides. However, it’s unclear how the reproducibility of phosphorylated peptide enrichment and dephosphorylation is as the variance across sample preparations often happens. It’s necessary to provide data to show reproducible enrichment and dephosphorylation using the protocol in the manuscript.
7. Line 210: What’s the source of kinases used in in vitro reactions? Are they purified endogenous kinases, recombinant full-length kinases, or just catalytic domain of kinases? The authors need to ensure that the identified phosphorated peptides from the in-vitro reactions are consistent with endogenous phosphorylated peptides. In other words, the kinases used in in vitro reactions will not over-phosphorylate peptides. This can be answered by comparing the results from in-vitro reactions with the phosphorylated peptide database, and in-vitro phosphorylation against peptides from the HeLa proteome without any enrichment.
8. Line 226 is not clear. Many factors (e.g., quantitation strategies, and instrument conditions) affect quantitation accuracy. Also, this no direct evidence indicating the reduced complexity would result in better quantitation accuracy. The author should consider the statement in a specific context.
9. Line 258-261: The authors need to cite references here.
10. Line 270: Less phosphorylated peptides from the motif-targeting approach, which is not consistent with the results in Figure 2B. The authors should provide a reasonable explanation.
11. Figures 3A and 3B are missing.
Author Response
Please see an attached file.

Reviewer 2 Report
This article by Ogata et. al. entitled “Motif-targeting phosphoproteome analysis of cancer cells for profiling kinase inhibitors” describes a method to enrich a subset of phosphopeptides with specific sequence motifs enabled by in vitro phosphorylation reaction using recombinant kinases. The authors have well described the method and showed the applicability of this method in profiling tyrosine phosphorylation status upon kinase inhibitor treatment. I recommend this manuscript to be published after the following concerns are addressed by the authors:
1. In the introduction section, it will be good to talk about the current standard methods for phosphoproteomic profiling such as Fe-IMAC and TiO2. In-depth analysis of phosphopeptides requires phosphopeptide enrichment from a fractionated sample, however, capturing the very low abundant phosphopeptides might still require additional methods as described in this article.
2. The following details are missing in the methods section:
a) The statistical analysis used in the article were not described
b) What FDR cut-offs were used to filter the database search results? It should be mentioned.
c) How the authors did inter-experimental normalization for phosphoproteomic profiling experiment upon kinase inhibitor treatment, as shown in the figure 5A.
3. The sentence “One of the major barriers…” on line 166 is misleading. LC-MS/MS analysis is biased towards any high abundant mass species and not necessarily phosphopeptides. Thus, enrichment of phosphopeptides is preferred, however, the global phosphoproteome analysis still misses very low abundant phosphopeptides due to wide dynamic range of phosphorylation.
4. Panel A and B in figure 3 are missing.
5. The authors mentioned about using triplicates for the analysis, however, only consolidated number was shown in the figure 2, 3 and supplementary 1A. All the results where multiple replicates were used should be shown individually for each replicate. These results will show the reproducibility of the described workflow.
6. Also, it needs to be clearly mentioned whether the replicates used in the study are process replicates or technical replicates for the LC-MS/MS analysis.
7. Did the authors note any difference in the identification of phosphopeptides with dephosphorylation step before LC-MS/MS analysis? If there is not major difference that authors observed, it could eliminate one more step from the workflow.
8. Line 301: “The significantly changed peptides were extracted by using ANOVA significance analysis of log- normalized intensities with the module in Perseus software”. The result of this analysis should be provided as a supplementary file.
9. Line 304: “1409 peptides that were regulated by at least one kinase inhibitor treatment”. What criteria was used to say that the peptides were regulated by kinase inhibitor treatment.
10. The authors should explain the discordant results between the conventional and motif-targeting workflow in figure 5B. I am wondering if the peptides identified in the motif-targeting workflow are identified in the conventional workflow.
11. The authors should report if the tyrosine phosphopeptides identified after in vitro kinase reaction with FYN, JAK3 and MER are already reported in the literature as substrates? A comparison to the public portals such as PhosphoSitePlus might be useful.
12. Enrichment of tyrosine phosphorylated peptides using pTyr-1000 kit is the current standard for quantitative profiling of tyrosine phosphorylated sites. Isn’t beneficial to do a global tyrosine phosphorylation analysis using pTyr-1000 kit instead of motif-targeting workflow as described in this article?
Author Response
Please see an attached file.

Round 2
Reviewer 1 Report
The authors have addressed all of my concerns in the revised version.